# MOBILE OBJECT REARRANGEMENT WITH LEARNED LOCALIZATION UNCERTAINTY

## ABSTRACT

Mobile object rearrangement (MOR) is a pivotal embodied AI task for a mobile agent to move objects to their target locations. While previous works rely on accurate pose information, we focus on scenarios where the agent needs to always localize both itself and the objects. This is challenging because accurate rearrangement depends on precise localization, yet localization in such a non-static environment is often disturbed by changes in the surroundings after rearrangement. To address this challenge, we first learn an effective representation for MOR only from sequential first-person view RGB images. It recurrently estimates agent and object poses, along with their associated uncertainties. With such uncertainty-aware localization as the input, we can then hierarchically train rearrangement policy networks for MOR. We develop and open source a simplified, yet challenging 3D MOR simulation environment to evaluate our method and relevant embodied AI baselines. Extensive comparisons reveal better performances of our method than baselines and the need for uncertainty estimation in our task.

## 1 INTRODUCTION

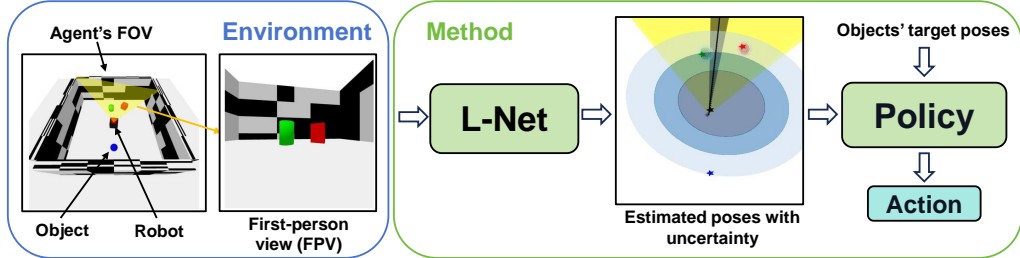

Figure 1: Overviews. *Left*: an overview of our MOR simulation environment. The robot endeavors to rearrange multiple primitive objects using solely a first-person view RGB image (FPV) as input. *Right*: an overview of our method. A localization network (L-Net) predicts object and robot poses with uncertainties from the FPV. The colored stars (★) indicate the ground-truth (GT) poses, and the colored contours show estimations of object and robot poses with their uncertainties parameterized as multivariate Gaussian distributions. The black straight line and the gray circular sector represent the GT and the estimated orientation (a Von Mises distribution) of the robot. Finally, the policy network leverages the uncertainty-aware estimations and the target object poses (directly given to the agent) to decide an action.

In the real world, both humans and animals possess the ability to rearrange objects in their surrounding environments to achieve specific objectives, such as cleaning a cluttered room or constructing nests and burrows. The development of engineered systems capable of performing such rearrangement tasks holds immense scientific and societal significance. Within the realm of embodied AI, the "rearrangement" tasks can be defined as the process of transitioning from an initial object configuration, on a tabletop or in a room, to a desired one through interactive actions (Batra et al., 2020a; Weihs et al., 2021b; King et al., 2016). In this paper, as shown in Figure 1, we focus on tackling this task using a mobile AI agent that needs to explicitly or implicitly solve a series of subtasks as Partially Observable Markov Decision Processes (POMDPs), including visual navigation, scene exploration, and sequential decision making.

We term our task as Mobile Object Rearrangement (MOR). The word "mobile" is used to highlight the need for localization in our task due to the movement of the mobile agent. This differs MOR from other popular rearrangement tasks that either use an internal GPS (Ehsani et al., 2021; Gu et al., 2022) or expose the ground-truth (GT) positions to the agent's policy (Weihs et al., 2021b; Ehsani et al., 2021; Yokoyama et al., 2023). While those tasks focus on rearrangement challenges in complex 3D scene understanding and reasoning, MOR focuses on another rearrangement challenge that stresses "no access to privileged information such as 3D models, ground-truth object poses, or perfect localization" as advocated by Batra et al. (2020a), calling for accurate *localization* and *navigation* to current and target object positions in the changing environment due to simultaneous *rearrangement*. This key challenge was studied by Han et al. (2023) in a simplified grid world. More realistically, MOR adopts visual inputs from 3D simulations that are easier to transfer to real-world robotics systems. Although MOR's scenes are simpler than other rearrangement environments like Habitat 2.0 (Szot et al., 2022a), it is enough to reveal the challenge while being fast enough for Deep Reinforcement Learning (DRL) to tackle this task with visual inputs.

The intuitive way to address the challenges outlined above is to decouple the representation learning (e.g., pose estimation from visual inputs) with policy learning (Stooke et al., 2021; Gadre et al., 2022; Han et al., 2023). Therefore, we develop a model capable of predicting generic poses (positions and orientations), encompassing both the agent and the objects in the environment. Additionally, this model has the ability to estimate uncertainty in these pose predictions. Then, following previous works (Szot et al., 2022a), we decompose the long-horizon rearrangement task into sub-tasks to reduce the timescale and make it more manageable. Accordingly, we divided our overall policy into three distinct components: two sub-policies (a pick policy and a drop policy) and a decision policy that outputs high-level actions (e.g., which object to pick up). This division makes our long-horizon rearrangement task easier to train and converge, and this separation can also benefit our model's scalability. Through our experiments, we have demonstrated that our model architecture and training pipeline design yield significant performance improvements over existing baseline models. Furthermore, our approach exhibits robust performance in scenarios with diverse environment configurations. We thus highlight our contributions as the following:

**1. Vision-only Rearrangement:** We propose a rearrangement task and a model without the input of depth maps, GPS, or GT positions required in previous works. Our model enables the agent to localize itself and the objects while exploring the environment and performing rearrangements.

**2. Uncertainty-aware Rearrangement:** We introduce a DRL-compatible localization uncertainty estimation into mobile rearrangement. Our experiments reveal a better and more stable policy learned thanks to the uncertainty estimation.

**3. Simultaneous Exploration and Rearrangement:** Our method allows more efficient and convenient rearrangement without prior exploration, while many previous works often require an exploration phase for mapping and scene understanding/memorization.

## 2 RELATED WORKS

### 2.1 EMBODIED AI TASKS

In recent years, we have witnessed a surge of interest in learning-based Embodied AI tasks. Various tasks have been proposed in this domain, including embodied question answering (Gordon et al., 2018; Das et al., 2018), instruction following (Anderson et al., 2018b; Shridhar et al., 2020; Ding et al., 2023), navigation towards objects (Batra et al., 2020b; Yang et al., 2018; Wortsman et al., 2019; Chaplot et al., 2020b; Huang et al., 2023), or towards a specific point (Anderson et al., 2018a; Wijmans et al., 2019; Ramakrishnan et al., 2020), decision making (Nottingham et al., 2023), scene exploration (Chen et al., 2019; Chaplot et al., 2020a; He et al., 2023b), object manipulation (Fan et al., 2018; Christen et al., 2022; 2023), and many others. In our work, the rearrangement task can be considered as a broader task that encompasses skills learned through aforementioned tasks, mainly including navigation towards an object and a point, scene exploration, and decision making.

## 2.2 MOBILE OBJECT REARRANGEMENT

Rearranging objects is a critical task for service robots, and much research has focused on moving objects from one location to another and placing them in a new position. Examples include the Habitat Rearrangement Challenge (Szot et al., 2022b) and the AI2-THOR Rearrangement Challenge (Weihs et al., 2021a). There is a rich literature on object rearrangement in robotics. Some of the works address the challenge in the context of the state of the objects being fully observed (Ben-Shahar & Rivlin, 1998; Stilman et al., 2007; King et al., 2016; Krontiris & Bekris, 2016; Yuan et al., 2018; Correll et al., 2016; Labbé et al., 2020). In contrast, there is also a rising interest in visual rearrangement (Batra et al., 2020a; Qureshi et al., 2021; Goyal et al., 2022; Trabucco et al., 2022; Wei et al., 2023), where the states of objects and the rearrangement goal are not directly observed. In these scenarios, the agent is presented with direct visual input, requiring it to extract relevant information from this visual data. However, a greater challenge arises when environmental factors, such as noise or external disturbances, disrupt the accuracy of an internal GPS system, particularly in confined indoor settings. In such cases, both the agent's position and the positions of objects become untraceable. Previous works (Weihs et al., 2021b; Gadre et al., 2022) have addressed this issue in situations where the rearrangement task has landmarks for the target positions or specific target locations, such as bookshelves or plates, that can serve as reference points discernible from the visual input. Our research also focuses on the visual rearrangement task, where GT positions of the agent and objects are not available. Moreover, we assume the absence of identifiable landmarks, as in some instances, individuals may prefer to specify a precise global position for the rearrangement objective. Consequently, our agent must possess the capability to extract a global understanding from its visual input encompassing both its own state and the surrounding environment. To tackle this challenge, we delegate the localization task to the agent itself, enhancing its stability in complementing its rearrangement capabilities.

## 2.3 PLANNING UNDER UNCERTAINTY

In real-world tasks or simulation tasks, there is often a significant amount of uncertainty due to partial observation. Most of the time, we cannot fully understand the entire environment because our observation is restricted. Therefore, various works have emerged to address the influence of uncertainty in different planning tasks. These works focus on different aspects, such as enhancing the agent's robustness and success probability under uncertainty (Luders et al., 2010; Liu & Ang, 2014; Tigas et al., 2019; Zhao et al., 2021; Pairet et al., 2021; He et al., 2023a), speeding up the agent's learning rate under uncertainty (Bry & Roy, 2011; Ho et al., 2023), or reducing uncertainty through agent actions (Wong et al., 2013; Xiao et al., 2019). In our work, we take a different perspective on these uncertainties which is potential to help the agent better understand the environment. Utilizing uncertainty has been proven effective in various tasks like object detection (Kendall & Gal, 2017; He & Wang, 2019), face alignment (Kumar et al., 2020), automatic driving (Feng et al., 2018; 2019; Su et al., 2023), and etc. In our work, we present a way to represent the uncertainty of the agent itself and manipulable objects in the rearrangement task. We have found that introducing these forms of uncertainty can help our agent complete the rearrangement tasks more efficiently and enhance the success rate.

## 3 THE MOBILE OBJECT REARRANGEMENT (MOR) TASK

### 3.1 TASK DEFINITION

In alignment with the task framework akin to visual room rearrangement (VRR) (Weihs et al., 2021b), we propose our MOR task which requires an agent to rearrange objects to recover a goal configuration from an initial state (shown in Figure 2(c)). In order to focus on studying the agent's capability of completing rearrangement tasks without the help of accurate position information, we streamline the scene complexity and object variety by confining the agent's interactions to simple primitives like cubes, spheres, etc., in a closed area. Specifically, we have 24 different primitives (labeled as integers $c \in \{0, 2, ...23\}$), resulting from the combination of four object categories and six unique colors, and we randomly choose $n$ of them in each episode. Meanwhile, each wall has a different checkerboard texture, randomly sampled from a fixed pool, for the agent to refer to localize itself. Therefore, *the wall texture and objects' category and color are changed every episode*, which makes our environment dynamic and the localization non-trivial for agents. Finally, our focus is

exclusively on rigid-body objects and abstracts away certain intricacies of the interaction physics between the agent and objects.

We formulate the MOR task as a partially observable Markov decision process (POMDP) problem which is defined as a tuple $\langle \mathcal{S}, \mathcal{A}, \mathcal{R}, \mathcal{T}, \mathcal{O} \rangle$. The state space $\mathcal{S}$ is a set of all possible poses of agent and objects, and state $s \in \mathcal{S}$ is denoted as a tuple $s = \langle s_{agent}, s_{objs} \rangle$, where $s_{agent}$ and $s_{objs}$ denote the state of the agent and multiple objects. Specifically, we let $s_{agent} = (x_{agnet}, y_{agent}, \theta) \in$ **SE**$(2)$, where $(x_{agnet}, y_{agent})$ and $\theta$ represent the position and orientation of the agent, respectively. Meanwhile, the objects' state is denoted as $s_{objs} = \{s_{obj_i} | i = 1, 2, ..., n\}$, wheres $s_{obj_i} = (x_i, y_i) \in \mathbb{R}^2$ and $(x_i, y_i)$ represents the position of object $i$. The agent's goal is to recover the target states of objects $s_{objs}^*$ from initial states $s_{objs}^0$ within a maximum number $N_{max}$ of steps.

As a POMDP problem, the agent can only perceive an RGB image by a mounted forward-facing camera as observation $o \in \mathcal{O}$, which makes the global state $s$ not directly accessible to the agent. At each time step $t$, the robot can take an action $a \in \mathcal{A}$ (details in Supplementary Section A), either moving, picking, or dropping an object in front of itself within a range, and gain reward $r \in \mathcal{R}(s^t, a, s^{t+1})$. Finally, the state transition model $\mathcal{T} : (s^t, a) \to s^{t+1}$ is added the Gaussian noise to the moving and rotating actions (see details in section A) to simulate the real-world scenarios where the motion control of the mobile robot is imperfect.

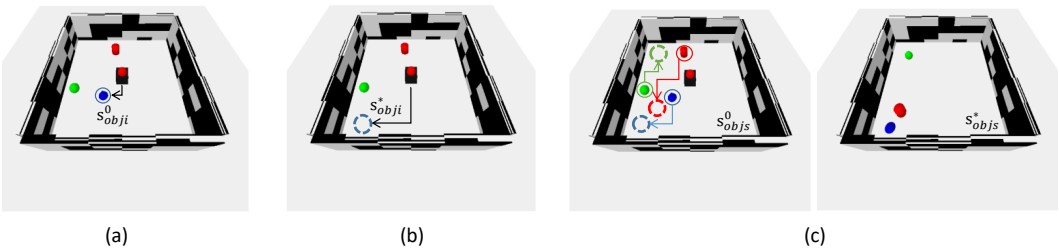

(a)  (b)  (c)

Figure 2: MOR task illustration. (a) **Pick**$(s_{obj_i}^0)$ task requires the agent to pick up an object $i$ at state $s_{obj_i}^0 = (x_i^0, y_i^0)$. (b) In the **Drop**$(s_{obj_i}^*)$ task, the agent needs to drop off the picked object which is collected in the agent's backpack (here, we remove it from the environment to indicate the picked status) to the target state $s_{obj_i}^* = (x_i^*, y^*)$. (c) The full rearrangement task needs the agent to reconfigure multiple objects from initial state $s_{objs}^0$ (left figure of (c)) to the target state $s_{objs}^*$ (right figure of (c)).

## 3.2 REARRANGEMENT TASK DECOMPOSITION

In this section, we decompose the entire MOR task into a series of subtasks. In other words, the completion of the entire task is achieved by successfully executing a sequence of pick and drop subtasks. As formulated by (Gu et al., 2022), each subtask is a sub-POMDP $\langle \mathcal{S}, \mathcal{A}_{sub}, \mathcal{R}_{sub}, \mathcal{T}, \mathcal{O} \rangle$ originated from the full rearrangement task.

**Pick**$(s_{obj_i}^0)$. For the pick task (shown in Figure 2(a)), the agent aims to pick up an object $i$, which is randomly chosen from $n$ objects as the target, at its initial state $s_{obj_i}^0 = (x_i^0, y_i^0)$. The agent is randomly spawned within the environment as the initial position at the start of each episode. The sensing model remains consistent with that of the full rearrangement task, wherein the agent is limited to accessing only an RGB image. The episode terminates when: (1) the agent executes the pick action (regardless of whether it picks up the target object, a wrong object, or nothing), (2) the agent cannot move due to obstruction, or (3) the game reaches the maximum episode horizon $N_{max} = 150$.

**Drop**$(s_{obj_i}^*)$. For the drop task (shown in Figure 2(b)), the agent is expected to drop off the hold object $i$ to its target state $s_{obj_i}^* = (x_i^*, y_i^*)$, which is directly given to the agent. At the beginning of each episode, the agent is placed at position $(x_i^0, y_i^0)$ of the object $i$. Then, object $i$ stays at the picked-up status until the agent drops it off. Similar to **Pick**$(s_{obj_i}^0)$, the episode terminates when: (1) the agent executes the drop action (no matter if it drops off object $i$ at the target or a wrong position), (2) the agent is obstructed, or (3) the number of steps reaches $N_{max} = 150$.

**Rearrangement task.** For the full rearrangement task (shown in Figure 2(c)), the agent needs to move objects from randomly initial state $s_{objs}^0$ to the target state $s_{objs}^*$. This is a long-horizon task which is composed of a sequence of **Pick**($s_{obj_i}^0$) and **Drop**($s_{obj_i}^*$) subtasks. The episode ends when: (1) the agent rearranges all objects back to where they belong, (2) the agent is obstructed, or (3) the game reaches the maximum episode horizon $N_{max} = 1000$.

### 3.3 Evaluation Metrics

To evaluate the agent's performance, we introduce two metrics for both pick and drop subtasks and adapt three metrics from VRR to quantify the entire rearrangement task. Before introducing the evaluation metrics for each task, it is essential to establish criteria for recognizing when an object can be considered as successfully rearranged to its target state. This is necessary because it is unreasonable to expect the agent to precisely position the object at its exact target location. Regarding this, we let $s_{obj_i} \approx s_{obj_i}^*$ represent the object $i$ is reconfigured to its target state, and we define that $s_{obj_i} \approx s_{obj_i}^*$ if, and only if, the euclidean distance $D(s_{obj_i}, s_{obj_i}^*)$ between $s_{obj_i}$ and $s_{obj_i}^*$ is less than 1. Next, we explain our metrics in detail.

**Pick&Drop subtasks metrics:**

**Success** - For subtask **Pick**($s_{obj_i}^0$) or **Drop**($s_{obj_i}^*$), **Success** equals 1 if, and only if, the agent pick up object $i$ or drop off the object $i$ at its target position $(x_i^*, y_i^*)$, otherwise it equals 0.

**Rearrangement task metrics:**

**1. Success** - For the full rearrangement task, **Success** equals 1 if, and only if, all objects are rearranged to the target positions – i.e., $s_{obj_i} \approx s_{obj_i}^*, \forall\, i$, otherwise it equals 0.

**2. % Fixed Strict (FS)** - Because the metric **Success** is too strict that it equals 0 if only one object is misplaced, we adapt the metric **FS** which measures the ratio of objects correctly rearranged to the total number of objects.

**3. % Energy Remaining (%E)** - For calculating this metric, we firstly let the energy function be defined as the summation of the Euclidean distance $D$ between each object's current position and its target position, $\sum_{i=1}^n D(s_{obj_i}, s_{obj_i}^*)$. Then, the metric **%E** equals the remaining energy when the episode ends divided by the initial energy - i.e., $\sum_{i=1}^n D(s_{obj_i}, s_{obj_i}^*) / \sum_{i=1}^n D(s_{obj_i}^0, s_{obj_i}^*)$.

## 4 Methods

In this section, we introduce our method (shown in Figure 3) for solving the MOR task. As we know localizing the manipulable objects and agent itself is crucial for the object rearrangement task, especially for our task where the GT poses of both agent and objects are not known, it is intuitive to involve a localization module for estimating pose information explicitly. However, the localization uncertainties exist due to the stochastic transition model of the environment and partial observation of the agent. To this end, as one of the major components of our pipeline, L-Net is designed to quantify the agent's and objects' poses with uncertainty by taking the raw image as input. Then, a hierarchical policy network takes the output from L-Net to make an action. In the following sections, we will explain each component in detail.

### 4.1 Localization Network (L-Net) for estimating pose with uncertainty

The L-Net is composed of two networks (shown in the top row of Figure 3). A PoseNet, denoted as $f_\Theta(o_t, c) \to \hat{s}^{ego,t}$, estimates state $\hat{s}^{ego,t} = \langle \hat{\mu}_{agent}^{G,t}, \hat{\Sigma}_{agent}^{G,t}, \hat{\theta}^{G,t}, \hat{\kappa}^{G,t} \{ \hat{\mu}_{obj_i}^{ego,t}, \hat{\Sigma}_{obj_i}^{ego,t} | i = 1, 2, ..., n \} \rangle$ of a single frame with observation $o_t$ and the category&color integer $c$ of each object. Here, $ego$ means the robot's egocentric coordinate and $G$ denotes the global coordinate. Then, an RNN model denoted as $g_\Theta(h_t, \hat{s}^{ego,t}) \to h_{t+1}, \hat{s}^{G,t}$, aggregates the state information over time to estimate the global state $\hat{s}^{G,t} = \langle \hat{\mu}_{agent}^{G,t}, \hat{\Sigma}_{agent}^{G,t}, \hat{\theta}^{G,t}, \hat{\kappa}^{G,t} \{ \hat{\mu}_{obj_i}^{G,t}, \hat{\Sigma}_{obj_i}^{G,t} | i = 1, 2, ..., n \} \rangle$.

Specifically, in PoseNet, a CNN is firstly used to extract features from $o_t$ at timestep $t$. Next, the category&color integer $c$ of each object is embedded into a vector $v$, which is then concatenated

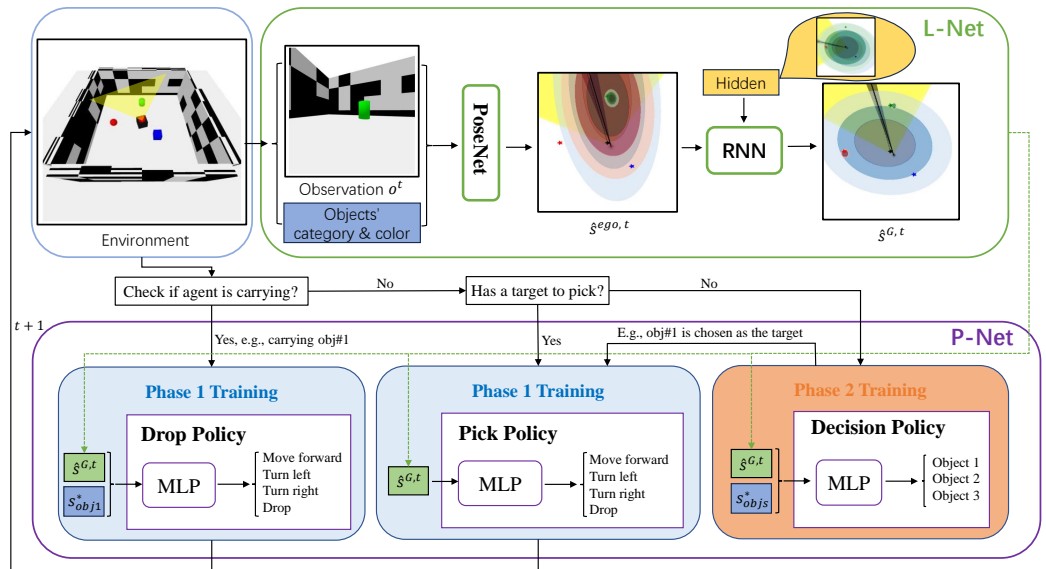

Figure 3: The pipeline of our method. The top row shows the localization network (L-net) that takes the observation $o$ and category&color $c$ of each object as input and estimates the agent's state. The bottom shows the hierarchical policy network (P-Net) that receives the information from the L-Net and outputs an action. During training, the drop and pick polices are trained separately in the phase I training. Subsequently, during Phase II training, the decision policy is trained while keeping the two previously trained sub-policies frozen.

with the image feature. Finally, an MLP is utilized to map the concatenated vector to the predicted state $\hat{s}^{ego,t}$ of the current frame. However, the PoseNet is constrained to predicting the partial state based on a single frame input, that makes it challenging to learn an effective control policy based on this limited information. As shown in the figure 3, the PoseNet assesses a multivariate Gaussian distribution for the green object, characterized by a minor variance, using the current $o_t$ as input. Conversely, the predicted distributions for the other two objects exhibit a more extensive spread, since PoseNet is not sure about their poses without perceiving them. Therefore, we introduce another RNN-based model to integrate the individual partial state over time to appraise the global state. Specifically, the hidden state $h_t$ which conveys the historical information and the output $\hat{s}^{ego,t}$ from PoseNet are fed into a RNN model to output the updated hidden state $h_{t+1}$ and estimate the global state $\hat{s}^{G,t}$. The detailed model architecture is presented in Supplementary Section 5.

These two components, the PoseNet $f_\Theta$ and the RNN model $g_\Theta$, are pre-trained separately via supervised learning. To accomplish this, we collect a dataset by allowing the agent to execute random actions within the simulation environment and storing the whole sequence of data, denoted as $\{M_k | k = 1, 2, .., N\}$. Each data point is defined as $M_k = (o^k, \mu_{agent}^k, \theta^k, \{\mu_{obj_i}^k | i = 1, 2, ..., n\})$, where $\mu_{agent}^k$, $\theta^k$ and $\mu_{obj_i}^k$ are the GT position and orientation of the agent and GT pose of object $i$, respectively. Here, we make the assumption that each estimated object's pose and the agent's position follow a multivariate Gaussian distribution, as illustrated in Supplementary Eq. 3. Meanwhile, we assume the estimated agent's orientation adheres to a Von Mises distribution (shown in Supplementary Eq. 4), and consider all of these distributions to be independent of each other. Then, we estimate the trainable parameter $\Theta$ of each network by minimizing the negative log-likelihood loss, defined as Eq. 5 in Supplementary, on a given dataset $\{M_k | k = 1, 2, .., N\}$.

## 4.2 HIERARCHICAL POLICIES FOR MOR

**Pick & Drop policy:** Since we decompose the full rearrangement task into two subtasks: **Pick**$(s_{obj_i}^0)$ and **Drop**$(s_{obj_i}^*)$ task, two sub-policies are trained through reinforcement learning method to tackle each of these subtasks. Here, we define the policy for **Pick**$(s_{obj_i}^0)$ task as $\pi_{pick}(\hat{s}^{G,t}, i)$ which takes the estimated global state $\hat{s}^{G,t}$ from L-Net and the order number of object

$i$ as input. The goal of $\pi_{pick}(\hat{s}^{G,t}, i)$ is to pick the object $i$ which is at state $s^0_{obj_i}$. For the **Drop**$(s^*_{obj_i})$ task, a corresponding policy $\pi_{drop}(\hat{s}^{G,t}, s^*_{obj_i})$ is learnt to drop the hold object at $s^*_{obj_i}$.

**Rearrangement policy:** The rearrangement policy is managed to decide high-level actions to chain up the sub-policies to complete the long-horizon MOR task hierarchically. This policy is defined as $\pi_{dec}(\hat{s}^{G,t}, \{s^*_{obj_i} | i = 1, 2, ..., n)\}$ which tasks the global state $\hat{s}^{G,t}$ and all obejcts' target state $\{s^*_{obj_i} | i = 1, 2, ..., n)\}$ as input to decide which object should be picked up when agent holds nothing. Then, the learned policy $\pi_{pick}(\hat{s}^{G,t}, i)$ will make a sequence of actions to pick up the chosen object. Next, the policy $\pi_{drop}(\hat{s}^{G,t}, s^*_{obj_i})$ drives the agent to drop off the held object to the target position. The full task is completed by performing a sequence of these sub-policies.

## 5 EXPERIMENTS

**Train and test protocol** - We collect 10,000 episodes of data by letting the agent execute random actions within the environment. The color and types of objects, and the wall texture are changed every episode. We split this dataset into train/test/valid by ratio 8:1:1 for training, testing, and evaluating the $f_\Theta$ and $g_\Theta$. Then, each policy network is trained by the PPO (Schulman et al., 2017) method for more than 1M steps with 20 CPU cores and 1 RTX8000 GPU, and is tested for 500 episodes for each task. All baselines and our method are trained in an environment with $n = 3$ manipulable objects.

**Model hyperparameter** - For the PoseNet $f_\Theta$, we use ResNet18 (He et al., 2016) to extract visual features from the RGB image input. Then, a three-layer MLP is used to map the feature and the category&color $c$ to the estimated poses and uncetainties of agent and objects. The LSTM is deployed for the RNN model $g_\Theta$. All sub-policies and the high-level rearrangement policy are five-layer MLP.

**Reward functions** - For each task, we define specific reward functions to guide the agent to learn an effective policy. For subtask **Pick**$(s^0_{obj_i})$, we define the reward function $r_{pick}(s^t, a, s^{t+1})$ as Eq. 1.

$$r_{pick}(s^t, a, s^{t+1}) = -\Delta D + 10\mathbb{I}_{success} - \mathbb{I}_{collision} \qquad (1)$$

Where $\Delta D$ is the change in the Euclidean distance $D_t$ between the agent's position $(x^t_{agent}, y^t_{agent})$ and the target object $i$'s position $(x_i, y_i)$ and $\Delta D = D_{t+1} - D_t$. $\mathbb{I}_{success}$ is the binary indicator if the agent successfully picks up the target object and $\mathbb{I}_{collision}$ indicates if agents collide with obstacles.

The task **Drop**$(s^*_{obj_i})$ uses a similar reward function as Eq. 1, but the $D$ measures the distance between the agent's position and the target position $(x^*_i, y^*_i)$ of object $i$. The reward function of the **full rearrangement task** is defined as Eq. 2. It is noted that we introduce a penalty of 0.01 to guide the agent to learn an efficient policy that minimizes the number of steps used for task completion.

$$r_{MOR}(s^t, a, s^{t+1}) = -0.01 + 10\mathbb{I}_{success} - \mathbb{I}_{collision} \qquad (2)$$

**Baselines** - Since FPV image is the only input from our MOR task, prior works that either need depth or GT agent's pose to build metric maps (Weihs et al., 2021b; Trabucco et al., 2022) cannot be directly deployed to this task. To this end, We adapted models from Embodied-CLIP (Khandelwal et al., 2022) that demand only RGB input, as our baselines. We train each baseline via the PPO algorithm with the same training steps and test it with the same protocol as we use for our model.

### 5.1 BASELINE RESULTS

After the initial tests, the EmbCLIP-based agent demonstrated a lack of success in all subtasks, as well as in the complete rearrangement task. This outcome is hypothesized to stem from the domain disparity between the natural images on which CLIP is trained and the synthetic texture data collected from our environment. Therefore, we only include ResNet50 and ResNet18 as our baselines' visual encoder and follow the same model design as the Embodied-Clip for these baselines (details in Supplementary Section A).

Table 1 shows the results of GT policy, baselines, and our method. Firstly, our method achieves a 1.6x performance than the ResNet50 baseline on the Pick$(s^0_{obj_i})$ task, and also performs comparable to the GT policy (uses GT poses of agent and objects as input) on this subtask. This indicates the explicitly estimated pose offers richer information than ResNet features. Then, taking the only RGB

and an object target pose $obj_i{}^*$ (in global coordinate) as input, baselines (ResNet + Img) fail to learn any effective policy on the $\text{Drop}(s^*_{obj_i})$ task. We postulate that the agent faces challenges in deriving a comprehensive global state representation, specifically in the form of a metric map, from a singular RGB input. Consequently, when presented with a target pose within global coordinates, and in the absence of any visual cues (e.g., landmarks) or relative pose information between the agent and the target position (a common practice in rearrangement tasks (Szot et al., 2021)), the agent struggles to acquire a meaningful policy due to the limited information.

To verify this hypothesis, we conduct two additional experiments: (1) ResNet+$\Delta$Pose - Agents are provided with a relative pose from itself to $obj_i{}^*$; (2) ResNet+Landmark - A landmark is placed at the target position (shown in Figure 4) to guide the agent to drop off the object correctly. From Table 1, we can see that baseline agents achieve the same level of performance as the GT policy and even surpass our method on the drop task with additional induction. Although this is not a fair comparison, the result reveals that the global state or a visual bias is necessary for the drop task. Hence, it further indicates that a decoupled L-Net for explicit global state estimation (agent's and objects' poses with uncertainties), used in our method, offers a reasonable performance (0.665) on this task.

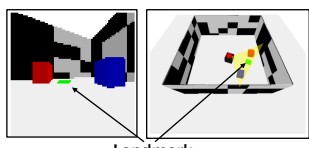

Figure 4: A green landmark, shown in left FPV and right BEV, is placed to guide the agent to drop the green object to the target position.

| | Pick($s^0_{obj_i}$) | Drop($s^*_{obj_i}$) | Rearrangement | | |
|---|---|---|---|---|---|
| | Success ↑ | Success ↑ | Success ↑ | % Fixed Strict (FS) ↑ | % Energy Remaining (%E) ↓ |
| Policy w/ GT poses | $0.795 \pm 0.030$ | $0.737 \pm 0.017$ | $0.476 \pm 0.016$ | $0.603 \pm 0.012$ | $0.442 \pm 0.011$ |
| ResNet18 + Img | | × | × | × | × |
| ResNet18 + $\Delta$Pose | $0.272 \pm 0.013$ | $\mathbf{0.724 \pm 0.028}^\dagger$ | $0.0132 \pm 0.007$ | $0.303 \pm 0.010$ | $0.749 \pm 0.007$ |
| ResNet18 + Landmark | | $0.671 \pm 0.029$ | $0.0036 \pm 0.002$ | $0.285 \pm 0.006$ | $0.770 \pm 0.011$ |
| ResNet50 + Img | | × | × | × | × |
| ResNet50 + $\Delta$Pose | $0.413 \pm 0.017$ | $0.717 \pm 0.020$ | $0.124 \pm 0.008$ | $0.350 \pm 0.026$ | $0.646 \pm 0.024$ |
| ResNet50 + Landmark | | $0.655 \pm 0.021$ | $0.0200 \pm 0.006$ | $0.318 \pm 0.009$ | $0.750 \pm 0.011$ |
| Ours | $\mathbf{0.707 \pm 0.020}$ | $0.665 \pm 0.007$ | $\mathbf{0.306 \pm 0.031}$ | $\mathbf{0.530 \pm 0.024}$ | $\mathbf{0.529 \pm 0.024}$ |

Table 1: Experiment results of GT policy, baselines, and our method. Each method is tested with 5 random seeds, and the mean and standard deviation across these seeds are reported for each metric. +Img means the agent takes only RGB image as input, +$\Delta$Pose means a relative pose from agent to target object position $s^*_{obj_i}$ is used for $\text{Drop}(s^*_{obj_i})$ task, and +Landmark denotes that a landmark is placed in the environment to indicate the target position for dropping (Figure 4). ×: Baseline with image input fails in the drop and rearrangement tasks. †: With relative pose as input, ResNet18 baseline outperforms our method on drop task. ↑/↓: higher is better/lower is better performance.

Last but not least, our method achieves better performance on the three metrics than baselines that rely on additional induction on the full rearrangement task. This reveals that decoupling a state estimation from policy learning is more effective than end-to-end RL policy training. Meanwhile, our model achieves comparable performance as the GT policy on the **FS** metric, demonstrating that the estimated poses with their uncertainties provide as complete information as the GT state. Notably, while our method achieves comparable performance to that of the GT policy in the pick and drop subtasks, our method's performance on the **Success** metric still falls short of the GT policy's. This discrepancy may be attributed to the accumulation of estimation errors over the task horizon.

## 5.2 Ablation study

We conduct a series of ablation studies on the effectiveness of uncertainty estimation and the impact of environment variations: (1) Comparison between policies trained with and without uncertainty estimation; (2) Introduction of an additional polygon-shaped environment to enhance task difficulty; (3) Incremental expansion of the rearrangement task's complexity by increasing the number of manipulable objects, ranging from two to five (details in Supplementary Section C).

**Uncertainty** - To validate the efficacy of leveraging estimated uncertainty for the rearrangement task, we conducted an ablation study in which we excluded the quantified variance information from the input provided to the policy network. Table 2 shows that the policy trained with uncertainty surpasses the policy without estimated uncertainty on all of the tasks, demonstrating the estimated uncertainty could offer the policy more induction to complete this rearrangement task where the GT

| | Pick($s^0_{obj_i}$) | Drop($s^*_{obj_i}$) | Rearrangement | | |
|---|---|---|---|---|---|
| | Success ↑ | Success ↑ | Success ↑ | % Fixed Strict (FS) ↑ | % Energy Remaining (%E) ↓ |
| Policy w/ GT poses | $0.795 \pm 0.030$ | $0.737 \pm 0.017$ | $0.476 \pm 0.016$ | $0.603 \pm 0.012$ | $0.442 \pm 0.011$ |
| Policy w/o uncertainty | $0.647 \pm 0.024$ | $0.649 \pm 0.021$ | $0.224 \pm 0.022$ | $0.474 \pm 0.020$ | $0.587 \pm 0.019$ |
| Ours (Policy + uncertainty) | $\mathbf{0.707 \pm 0.020}$ | $\mathbf{0.665 \pm 0.007}$ | $\mathbf{0.306 \pm 0.031}$ | $\mathbf{0.530 \pm 0.024}$ | $\mathbf{0.529 \pm 0.024}$ |

Table 2: Comparison between policies trained with and without uncertainty estimation. The estimated uncertainty offers a better performance of our method than the uncertainty unknown policy on both subtasks and the full rearrangement task.

pose of the agent is unknown or agent only has partial observation of the global state. An interesting behavior can be observed in the demo video (see supplementary material), where the agent tries to rotate itself at the starting position to decrease the uncertainty of each object before moving towards objects. This indicates the policy does leverage the uncertainty information for better rearrangement performance. Therefore, this insight encourages the following works to think about incorporating uncertainty into policy learning for embodied AI tasks.

**Environment variation** - We incorporate an irregular polygon-shaped environment (as shown in Figure 5) into our study as a means of introducing environmental variation where the localization would be more challenging. For this ablation study, we retrain the L-Net with the data collected from this new environment, and then we test our method with/without fine-tuning the policy on the new environment. Here, we use the ResNet50+ΔPose which achieves the best baseline performance on the original tasks, as the baseline. Table 3 indicates that with only retrained L-Net on this new environment, our policy achieves significantly better performance (mean 0.538 vs. 0.224) on the pick with zero-shot transfer. Without the need for retraining RL policy, which demands tremendous data, reveals the advantage of decoupling state estimation from policy learning. Meanwhile, fine-tuning the policy is as expected to offer better performance than baselines and zero-shot transfer policy on all tasks.

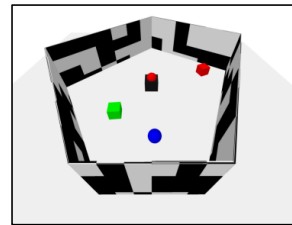

Figure 5: Illustration of the polygon-shaped environment where the agent has the same observation and actions as the original environment.

| | Pick($s^0_{obj_i}$) | Drop($s^*_{obj_i}$) | Rearrangement | | |
|---|---|---|---|---|---|
| | Success ↑ | Success ↑ | Success ↑ | % Fixed Strict (FS) ↑ | % Energy Remaining (%E) ↓ |
| ResNet50 + ΔPose | $0.224 \pm 0.019$ | $0.725 \pm 0.015^{\dagger}$ | $0.0592 \pm 0.044$ | $0.302 \pm 0.018$ | $0.732 \pm 0.017$ |
| W/o fine tuning | $0.538 \pm 0.020$ | $0.562 \pm 0.020$ | $0.0668 \pm 0.015$ | $0.297 \pm 0.013$ | $0.700 \pm 0.014$ |
| Fine tuning | $\mathbf{0.545 \pm 0.018}$ | $\mathbf{0.727 \pm 0.014}$ | $\mathbf{0.211 \pm 0.008}$ | $\mathbf{0.448 \pm 0.013}$ | $\mathbf{0.571 \pm 0.011}$ |

Table 3: Experiment results of our method, with/without fine-tuning, and the baseline on the rearrangement task in the polygon-shaped environment. Without fine-tuning, our method shows surpassing performance than the baseline. †: This baseline relies on the inclusion of additional relative pose information as input, which consequently renders the results not directly comparable.

## 6  CONCLUSION

We present a novel approach for the MOR task where the GT agent's and objects' poses are unknown. Our approach comprises a state estimation module designed to explicitly assess the poses of both the agent and manipulable objects, along with their associated uncertainties. Additionally, a policy net that leverages the quantified uncertainties to effectively solve the rearrangement task. Our method demonstrates surpassing performance than relevant embodied AI baselines on most of tasks. Furthermore, it exhibits reasonable behavior even in more complex scenarios, such as irregularly shaped environments. While our work abstracts certain irrelevant environmental dynamics and complex object interactions from the task, it provides valuable insights into the potential for leveraging uncertainty in policy learning for embodied AI tasks. This encourages further exploration of how uncertainty, stemming from partial observations and dynamic environments, can be integrated into policy learning.

## 7 REPRODUCIBILITY STATEMENT

To make all our experiment results reproducible, we submit codes with hyperparameters used for each experiment in the supplementary material. We also provide an instruction on how to use our code to reproduce the experiment results.

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

equation

## A    EXPERIMENT

**Action space** - We follow the similar action space definition of VRR task (Weihs et al., 2021b) which uses abstracted actions without considering complex interaction between the agent and the environment. The action space is defined as: $a \in \mathcal{A} = \{$MOVE, TURNLEFT, TURNRIGHT, PICK, DROP$\}$. MOVE results in agent moving forward with distance $d \sim \mathcal{N}(0.15m, 0.05^2)$, TURNLEFT and TURN-RIGHT make agent rotate $\theta \sim \mathcal{N}(10°, 1^2)$. It should be noted that *the two distributions (neither the parameters nor the types) are not known to the policy net.* PICK results in the agent picking up a visible object if the agent is not holding an object and is close to the object (within 1m). DROP makes the agent release the held object to the straight ahead of itself with 1.5m. It is noted that action DROP is removed in the subtask **Pick**$(s_{obj_i}^0)$, and action PICK is removed in subtask **Drop**$(s_{obj_i}^*)$.

**Baseline architecture** - Specifically, the CNN backbone extracts visual features from RGB input and a linear layer is used to embed category&color $c$ of the target object to high-dimension latent vector. Then, the flattened image feature, concatenated with the target latent vector, is fed to a MLP-based policy network to output an action. It is noted that the RNN-based policy used in Embodied-Clip is not deployed for reducing training cost in our baseline, since we find the performance of RNN-based is same as MLP-based policy in this task.

## B    METHOD

Here, we assume each estimated object's pose and agent's position to be a multivariate Gaussian distribution (shown in Eq. 3) with mean $\hat{\mu}$ and covariance $\hat{\Sigma}$. Specifically, the agent's position is assumed to follow a multivariate Gaussian distribution with mean $\hat{\mu}_{agent} = (\hat{x}_{agent}, \hat{y}_{agent})$ and covariance $\hat{\Sigma}_{agent}$. Meanwhile, it also appraises the pose of each object which also follows the multivariate Gaussian distribution with mean and covariance $\{\hat{\mu}_{obj_i} = (\hat{x}_i, \hat{y}_i), \hat{\Sigma}_{obj_i} | i = 1, 2, ..., n\}$. Then, we assume the estimated agent's orientation as a Von Mises distribution with mean $\hat{\theta}$ and concentration $\hat{\kappa}$ (shown in Eq. 4), and all of them to be independent.

$$P(\overline{\mu}|\hat{\mu}, \hat{\Sigma}) = \frac{1}{\sqrt{2\pi|\hat{\Sigma}|}} \exp\left(-\frac{(\overline{\mu} - \hat{\mu})^T \hat{\Sigma}^{-1}(\overline{\mu} - \hat{\mu})}{2}\right) \tag{3}$$

Where $\overline{\mu}$ is a possible pose sampled from the distribution. We use Cholesky decomposition (Kumar et al., 2020) to make sure covariance matrix $\hat{\Sigma}$ is symmetric positive definite in practice.

$$Q(\overline{\theta}|\hat{\theta}, \hat{\kappa}) = \frac{\exp(\hat{\kappa}\cos(\overline{\theta} - \hat{\theta}))}{2\pi I_0(\hat{\kappa})} \tag{4}$$

Where $\overline{\theta}$ is a possible orientation, $\hat{\kappa}$ is the concentration , and $I_0$ is the modified Bessel function of the first kind of order 0.

The total loss is defined as a summation of negative log-likelihood of each independent distribution, as shown in Eq. 5.

$$L = L_{agent} + L_{objs} = -\log P(\mu_{agent}|\hat{\mu}_{agent}, \hat{\Sigma}_{agent}) - \log Q(\theta|\hat{\theta}, \hat{\kappa})$$
$$- \sum_{i=1}^{n} \log P(\mu_{obj_i}|\hat{\mu}_{obj_i}, \hat{\Sigma}_{obj_i}) \tag{5}$$

## C    ABLATION STUDY ON THE NUMBER OF OBJECTS

We examine the performance of our method through a task of increasing complexity, wherein the number of objects increases. From Table 4, we observe a degrading performance on each task with the number of manipulable objects increasing. This decline can be attributed to the accrual of estimation errors, which become more pronounced as the task horizon expands (more object needed to be rearranged). However, our method achieves reasonable performance when number of objects equals four (e.g., success 0.089 means the agent can perfectly solve 9 of 100 games).

| Num. of | Pick($s_{obj_i}^0$) | Drop($s_{obj_i}^*$) | Rearrangement | | |
| objects | Success ↑ | Success ↑ | Success ↑ | % Fixed Strict (FS) ↑ | % Energy Remaining (%E) ↓ |
|---|---|---|---|---|---|
| 1 | $0.895 \pm 0.015$ | $0.889 \pm 0.005$ | $0.939 \pm 0.010$ | $0.939 \pm 0.010$ | $0.153 \pm 0.009$ |
| 2 | $0.752 \pm 0.013$ | $0.717 \pm 0.018$ | $0.592 \pm 0.008$ | $0.716 \pm 0.007$ | $0.355 \pm 0.006$ |
| 3 | $0.621 \pm 0.019$ | $0.604 \pm 0.020$ | $0.263 \pm 0.024$ | $0.491 \pm 0.017$ | $0.558 \pm 0.018$ |
| 4 | $0.515 \pm 0.024$ | $0.444 \pm 0.029$ | $0.0892 \pm 0.008$ | $0.325 \pm 0.008$ | $0.709 \pm 0.006$ |
| 5 | $0.294 \pm 0.014$ | $0.330 \pm 0.020$ | $0.0188 \pm 0.006$ | $0.206 \pm 0.007$ | $0.814 \pm 0.006$ |

Table 4: Ablation study on a task with incremental number of objects. Generally, the performance decreases as the number of objects increases, demenstraing the increasing task complexity. However, our method can still achieves reasonable performance with 0.089 success (perfectly complete 9 of 100 games).

## D  REAL WORLD EXPERIMENT

In this section, we show our developed hardware platform for MOR task. This hardware area shares similar configuration as our simulation environment, e.g., checkerboard wall texture, motion noise, and colored primitives. It should be noted that this hardware experiment showcases the results of policy with GT poses of agent and objects as input. Specifically, these GT information are acquired from a top-down view camera system. The policy is zero-shot transfer to the hardware experiment without any fine tuning. Next, we describe the hardware platform in details.

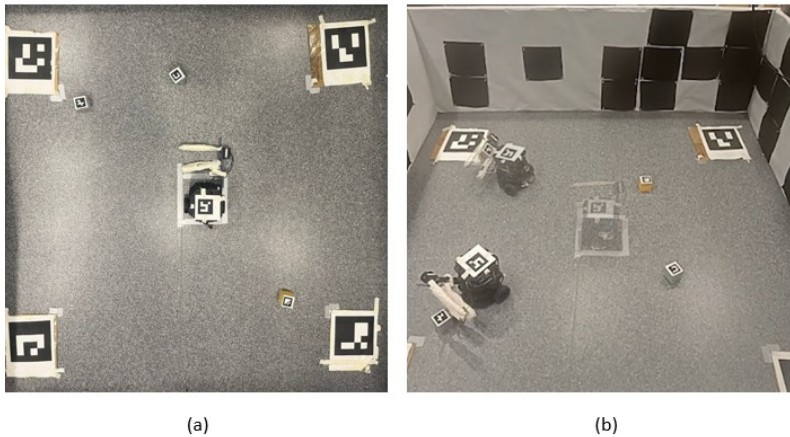

(a)                                          (b)

Figure D.1: (a) Top down view of the environment with agent in the center and three cubes distributed in the environment. (b) The agent executes the actions provided by the policy (trained in the simulation). It first orients itself before proceeding toward the target object, picks it up, and orients itself again before proceeding towards the target pose to place the object.

The proposed framework was tested in a dedicated hardware environment where TurtleBot - Burger was used as the agent. Notably, this setup had two cameras to facilitate the functionality. The top camera gave a top down view of the environment (as shown in Figure D.1(a)), enabling the determination of the pose of the agent and the objects in the environment. This pose information was determined by tracking the ArUco tags attached to the agent, the cubes, and the corners of the environment.

The term "action" indicates the mechanism by which the agent transitions between states of the environment. The actions included forward (advancement by 0.15 m), left turn (10 degrees leftward turn), right turn (10 degrees rightward turn), pick, and place actions, respectively. The real world was designed and built to closely resemble the simulation environment so as to transfer the policy with minimal adaptations.

There were a total of three objects in the environment whose predetermined target pose was provided as inputs to the model. Leveraging the pose data collected from the top-down view camera, the model provided the corresponding action to be executed at that instant in the real world. This process iterated until all the objects were placed at the target pose.

From the real-world experiment, we can see that our policy can be directly deployed into the real world without any fine tuning, and the video (in supplementary material) showcases a success rearrangement by the robots. In the future work, we will test our policy with only the FPV RGB image as input in this real world environment.

