# OpenReview forum: "Mobile Object Rearrangement with Learned Localization Uncertainty"
_ICLR.cc/2024/Conference — Submitted to ICLR 2024_

### Official Review · Reviewer_CMyF · 2023-10-30

**Soundness:** 3 good
**Presentation:** 3 good
**Contribution:** 3 good
**Rating:** 5
**Confidence:** 3

**Summary:**

The paper focuses on mobile object rearrangement (MoR) and the scenarios where the localization is imperfect. To this end, the authors propose to use a recurrent network (L-Net) to estimate agent and object poses with uncertainties (following Gaussian distributions), and train a policy network (P-Net) based on estimated poses. The authors compare the proposed method with baselines on a simulation benchmark, and show some qualitative results in the real world.

**Strengths:**

1. The paper studies the scenarios where the localization is imperfect, which is an under-explored aspect of prior works (e.g., Habitat 2.0, Multi-skill Mobile Manipulation for Object Rearrangement).

**Weaknesses:**

1. Missing baselines or explanations about related works. Can the authors explain why Neural SLAM [1] is not included as a baseline? In NeuralSLAM, the NeuralSLAM module estimates the 2D map and relative pose as what the L-Net does in this paper, and a global policy is trained to explore the map but can be adapted to achieve arrangement tasks as what the P-Net does in this paper. Besides, [2] showcases that using explicitly estimated poses can be helpful for learning mobile manipulation.


[1] Chaplot, Devendra Singh, et al. "Learning To Explore Using Active Neural SLAM." International Conference on Learning Representations. 2019.
[2] Cong Wang, Qifeng Zhang, Qiyan Tian, Shuo Li, Xiaohui Wang, David Lane, Yvan Petillot, and Sen Wang. Learning mobile manipulation through deep reinforcement learning. Sensors, 20(3):939, 2020.

**Questions:**

1. Does the "global coordinate" in Sec 4.1 mean "episodic coordinate"? For example, the initial agent position is considered (0, 0).
2. How do the authors acquire "relative pose" for the baseline "ResNet + $\Delta$Pose"? And why are results for these baselines missing for "Pick"?
3. Is the recurrent network used for baselines like ResNet + Img and ResNet + Landmark?
4. Can the authors qualitatively and quantitatively show how well the estimated poses are?

---

> ### Author Response · Authors · 2023-11-22
> **Neural SLAM is not suitable for MOR task.**
>
> Thank you for the constructive feedback and suggestions.
>
> 1. “why Neural SLAM [1] is not included as a baseline?”
>
> There are two primary reasons why neural SLAM may not be suitable for our task. Firstly, the dynamic nature of our environment, where object positions frequently change due to rearrangement actions, contradicts the static environment assumption inherent in neural SLAM. While in room environments like Habitat or AI2Thor, a slight change in the pose of a small object may still allow SLAM-based methods to be effective, our MOR environment simulates scenarios such as a warehouse where the substantial position change of large objects dramatically influences the scene, rendering the environment highly dynamic. Therefore, the application of neural SLAM in the MOR context becomes challenging.
>
> Secondly, the additional pose sensor used in neural SLAM is not involved in the MOR task. While it's possible to include this pose sensor in our simulation, doing so would make the results not directly comparable to ours.
>
> 2. “using explicitly estimated poses can be helpful for learning mobile manipulation”
>
> We appreciate the reviewer for recommending this paper, as it aligns with our approach of explicitly estimating poses for rearrangement tasks. However, our unique contribution lies in the exploration of additional pose uncertainties, demonstrating that they provide the agent with more information than poses alone, facilitating the learning of an effective rearrangement policy. This assertion is substantiated by our ablation study on uncertainty, as detailed in the experiment section. Our work extends beyond the mere estimation of poses by emphasizing the importance of uncertainty in enhancing the agent's performance in rearrangement tasks.
>
> 3. “Does the "global coordinate" in Sec 4.1 mean "episodic coordinate"? For example, the initial agent position is considered (0, 0).”
>
> The global coordinate is corresponding to the world coordinate where the agent is always reset at the center of a 12x12 size environment with position (6,6).
>
> 4. “How do the authors acquire "relative pose" for the baseline "ResNet + Pose"? And why are results for these baselines missing for "Pick"?”
>
> Cause we know the GT agent position and objects’ positions from the simulation environment. So we can easily compute the relative position between agent and objects. The reason why we provide the relative position to the Drop task is that the baselines fail to learn any policy by only RGB input. Therefore, we give the baselines more information to learn a reasonable policy. However, in the pick task, the baselines can already learn a reasonable policy. Therefore the relative pose is not involved in the pick task.
>
> 5. “Is the recurrent network used for baselines like ResNet + Img and ResNet + Landmark?”
>
> No, we do not use recurrent network for the baselines. As we stated in the experiment section, we find that the baselines have the same level of performance with or without recurrent networks. Therefore, for training efficiency, we don’t use recurrent networks for baselines.
>
> 6. “Can the authors qualitatively and quantitatively show how well the estimated poses are?”
>
> The demo videos in the supplementary showcases that when the objects are not in the view of the agent, the uncertainties of these objects are large. As the agent perceives the objects, the uncertainties drop to near zero. This qualitatively demonstrates the L-Net could effectively represent the uncertainty. For the quantitative results, we could not think about a way to quantify how well the L-Net could estimate the uncertainty. Following some related works like [1] which uses downstream task performance to quantitatively evaluate the uncertainties, we could use the performance of downstream rearrangement task to verify the effectiveness of L-Net for the uncertainty estimation quantitatively.
>
> [1] Su, Sanbao, et al. "Uncertainty quantification of collaborative detection for self-driving." 2023 IEEE International Conference on Robotics and Automation (ICRA). IEEE, 2023.

---

### Official Review · Reviewer_CAZo · 2023-10-31

**Soundness:** 3 good
**Presentation:** 3 good
**Contribution:** 2 fair
**Rating:** 5
**Confidence:** 3

**Summary:**

This paper proposes a variation of the Embodied AI rearrangement task, that drops the assumption of ground-truth agent pose and perfect localization of the objects during the task. The proposed approach first learns to localize jointly both the agent and the objects in the environment and employs a hierarchical pick and drop policy for executing the task.

**Strengths:**

The paper is easy to follow with the proposed task and methodology are clearly explained. The separate training of the pick and drop policies makes sense, I would be curious to see the performance improvement compared to a single training stage of the proposed hierarchical policy network.

**Weaknesses:**

My main issue with this work are the chosen simulation environments which are visually and structurally trivial with primitive objects. The paper mentions that the main challenge of the object rearrangement task is precise localization, but I fail to see how the localization of either the agent or the objects is a challenge in these environments. Have the authors tried classical visual odometry or monocular slam with object detection to register the objects in a map? The proposed L-Net seems kind of an overkill in this setup. The authors also mention that partial observability and their choice of using a single first-person RGB view makes the problem harder. The environment is small enough without obstacles that it can be fully observed in very few views, while including an RGB-D sensor is a fair assumption in robotic settings.

Even in cases where the localization is a challenge, it is an orthogonal problem to the re-arrangement task. I am expecting that the planning is more of a bottleneck here due to the non-trivial sequence of actions that need to be decided (i.e., moving a certain object first might lead to a state where an optimal solution is not possible anymore). This is also recognized as the main challenge of this problem by Batra et al. (2020a).

**Questions:**

--

---

> ### Author Response · Authors · 2023-11-22
> **MOR is still non-trivial because of the highly dynamic environment.**
>
> Thank you for the constructive feedback.
>
> 1. “My main issue with this work are the chosen simulation environments which are visually and structurally trivial with primitive objects.’
>
> We acknowledge that our simulation environment abstracts away certain irrelevant environmental dynamics, particularly the physical interactions between the agent and objects. However, despite this abstraction, the dynamics present in our simulation environment, such as changing wall textures, object types, colors in each episode, and introduced motion uncertainty, render the MOR task non-trivial for baseline methods and even challenging for our proposed approach, as evidenced by the experiment results.
>
> Furthermore, the constant changes in object positions due to rearrangement actions pose a significant challenge to localization. The highly dynamic nature of the environment also renders most static assumptions of classic visual odometry and SLAM methods invalid.
>
> Our L-Net is designed as a straightforward recurrent CNN that adeptly estimates both the agent's and objects' poses, along with their associated uncertainties, within the Manipulation and Object Rearrangement (MOR) task. We believe that this L-Net is a proper way instead of an overkill for the rearrangement task.
>
> Certainly, the acknowledgment that our environment is indeed partially observable is crucial, given the limited field of view (FOV) of 90 degrees, preventing the agent from fully observing the entire environment in a single step. As highlighted by the reviewer, the agents need a few steps to fully see the environment.
>
> We agree that the RGB-D sensor could be involved in MOR. However, it should be noted that knowing the depth information does not equal perfect localization. The highly dynamic fact in the MOR makes localization challenging even with depth information given.
>
> 2. “Localization is an orthogonal problem to the re-arrangement task”
>
> We concur with the acknowledgment that learning an optimal rearrangement policy is a pivotal research direction for tasks of this nature. However, we believe that effective localization serves as the foundational building block for successful rearrangement. The ability to precisely localize the agent and identify manipulable objects is crucial, representing key components in the overall challenge of acquiring and learning an optimal rearrangement policy.

---

### Official Review · Reviewer_XWmC · 2023-11-02

**Soundness:** 3 good
**Presentation:** 2 fair
**Contribution:** 2 fair
**Rating:** 3
**Confidence:** 4

**Summary:**

This paper studies the problem of geometric goal object rearrangement in a setting where the agent does not have ground-truth GPS+Compass. Towards this end, the authors propose a simplified environment to study this problem.

They propose a policy where agent tracking is decoupled from navigation/manipulation. The task itself is broken down into a high-level controller policy that selects which object to rearrange next, a pick policy, and a drop policy. Their proposed method is shown to outperform various baselines.

**Strengths:**

This paper studies an important problem
Baselines and ablations for the rearrangement policy are well thought-out.
The reviewer found the landmark experiment to be quite interesting.

**Weaknesses:**

The environment used is not visually realistic and does not support their claim that "MOR adopts visual inputs from 3D simulations that are easier to transfer to real-world robotics systems." The environment itself is a simple convex shape with no obstacles or visual features besides a black-and-white checkerboard pattern on the walls. The reviewer is concerned that these two extreme simplifications in the environment mean that any conclusions drawn won't transfer to more realistic settings. This concern is reinforced by the result that CLIP features were not appropriate for this environment.

Additional baselines are needed for L-NET. The embodied AI literature has a large number of tracking methods that were developed for PointGoal navigation, see Partsey et al 2022 for one example.

The name "Mobile Object Rearrangement (MOR)" is confusing as the word mobile could also be used to describe the agent being mobile and differentiate the task from table-top rearrangement.

The figure 3 caption is very long and has key method details. Method details should not be in figure captions as the reader does not know when they are supposed to read these details.



## References

Partsey et al 2022, Is Mapping Necessary for Realistic PointGoal Navigation?, CVPR 2022

**Questions:**

1. Is the agent always initialized in the center of the environment?
2. What is the advantage of using a distribution for agent/object pose instead of using regression?
3. Why is it necessary to predict object position? Since the object position is given relative the agent position at the start of the episode, the object's location with respect to the agent's position can always be computed given the agent's pose.

---

> ### Author Response · Authors · 2023-11-22
> **The simplified environment makes it easier to zero-shot transfer the trained policy to the real world.**
>
> Thank you for the constructive feedback.
>
> 1. “The reviewer is concerned that these two extreme simplifications in the environment mean that any conclusions drawn won't transfer to more realistic settings.”
>
> We agree that our simulation environment is a simplified toy-like environment. The intuition of conducting all experiments in this simplified, yet challenging environment is rooted in the belief that the environmental setups can be readily transferred to real-world scenarios, as demonstrated in the supplementary material. The showcased real-world demonstration illustrates the zero-shot transferability of the trained policy net from simulation to reality. While we acknowledge the importance of eventually working in a more photorealistic environment, such as Habitat, for greater relevance to the broader robotic community, we recognize this as a facet of our future work. The current emphasis on a simplified environment serves as a foundational step, demonstrating the feasibility of transferring policies from simulation to real-world scenarios.
>
> We acknowledge the noted failure of the CLIP-agent in our tasks, primarily attributed to the domain gap between our simplified environment and a more realistic setting. It is important to underscore that the primary contribution of this paper does not lie in the feature extraction domain. Rather, the core innovation resides in our proposed solution, which can be readily adapted to a more photorealistic environment by simply substituting the visual feature extraction module, such as CLIP, with a more suitable alternative. This flexibility in adapting the proposed solution underscores its robustness and applicability across varying environments.
>
> 2. “Additional baselines are needed for L-NET.”
>
> We express our gratitude to the reviewers for suggesting related papers. However, it is important to note that our paper specifically focuses on the rearrangement task, as opposed to navigation, with L-Net serving as just one component of our proposed method. Given this focus, we believe it may not be necessary to compare our work directly with navigation baselines.
>
> Furthermore, we would like to highlight another contribution of our paper, the explicit estimation of uncertainty and its beneficial impact on rearrangement performance. This aspect sets our work apart, as many navigation papers typically do not address uncertainty explicitly.
>
> 3. “The name "Mobile Object Rearrangement (MOR)" is confusing.”
>
> We thank the reviewer for pointing this out. The word “mobile” refers to the agent being mobile. As we stated in the abstract, Mobile object rearrangement (MOR) stands for the embodied AI task for a mobile agent to move objects to their target locations.
>
> 4. “The figure 3 caption is very long and has key method details.”
>
> Thank you for your comments. We have shorten the caption of Figure 3 and moved the details to the supplementary.
>
> 5. “Is the agent always initialized in the center of the environment?”
>
> Yes, the agent is always reset at the center of the environment.
>
> 6. “What is the advantage of using a distribution for agent/object pose instead of using regression”
>
> As shown in the ablation study on the uncertainty we conducted, the estimated uncertainty could offer induction for the agent to complete the rearrangement. For example, the agent will choose to explore the area with larger object position uncertainties to better capture the objects and the agent will choose to decrease its pose uncertainty to better localize itself. This behavior could be found in the videos in the supplementary material and discussed in the experiment section. By contrast, the simple pose regress could not offer such information.
>
> 7. “Why is it necessary to predict object position?”
>
> Actually, the object's position is not known to the agent and we do not give the relative position of the objects to the agent in MOR. In the experiment section, because the baselines fail to find the objects in the Drop task, we offer the relative poses to give more information. This makes the baseline results not directly comparable to our method. However, all other experiments do not have the relative poses as input.

---

### Official Review · Reviewer_ZaeQ · 2023-11-06

**Soundness:** 2 fair
**Presentation:** 2 fair
**Contribution:** 1 poor
**Rating:** 3
**Confidence:** 4

**Summary:**

This paper proposes Mobile Object Rearrangement (MOR), in which the agent is tasked with recovering a goal configuration from an initial state without the use of depth maps, GPS, or ground truth positions (GT positions). The authors introduce a modular approach that begins by training an L-Net to estimate poses with uncertainty, followed by learning a rearrangement policy through Hierarchical Reinforcement Learning (RL). The proposed method demonstrates superior performance compared to the baselines in the MOR tasks.

**Strengths:**

1. This paper addresses an important problem in visual object rearrangement: learning a policy that does not rely on privileged information as input.
2. The introduction is well-written.

**Weaknesses:**

1. (Motivation) Why couldn't the agent utilize depth information and conduct simultaneous localization and mapping (SLAM)? It doesn't seem necessary to completely forgo the use of internal GPS and depth sensors.

2. (Method) The proposed method heavily relies on the localization network (L-Net). However, training the L-Net necessitates extensive pre-training (e.g., 10,000 episodes in the proposed simplified environment), raising questions about its suitability for deployment in more complex environments like Habitat, AI2Thor, or the real world.

3. (Evaluation) The evaluation of the proposed method takes place in toy-like environments, and real-world experiments assume ground-truth poses of agents and objects.

4. (Writing) I recommend that the authors enhance the clarity of the writing in Sections 3 and 4. These sections currently contain an overwhelming amount of details and lack structured and logically coherent expressions.

**Questions:**

How does the agent perceive the target poses of the objects? Does it estimate these poses from another unshuffled environment?

---

> ### Author Response · Authors · 2023-11-22
> **Knowing depth or conducting SLAM does not equal perfect localization**
>
> Thank you for the constructive feedback.
>
> 1. “Why couldn't the agent utilize depth information and conduct simultaneous localization and mapping (SLAM)? It doesn't seem necessary to completely forgo the use of internal GPS and depth sensors.”
>
> We acknowledge the potential involvement of other sensors, such as internal GPS and depth cameras, in facilitating simultaneous localization and mapping (SLAM)-type techniques within our MOR task. However, it should be noted that possessing depth information or employing SLAM does not equate to achieving perfect localization. This paper concentrates on the challenge of developing an effective rearrangement policy without assuming perfect localization knowledge.
>
> As illustrated in the experimental section, our proposed L-Net demonstrates the ability to effectively estimate the poses of both the agent and objects, along with their associated uncertainties. Notably, these uncertainties play a pivotal role in enhancing the performance of MOR. In contrast, SLAM methods lack the capability to provide such uncertainty information for informing the rearrangement policy. Furthermore, the static environment assumption becomes invalid due to the dynamic nature of rearrangement actions, particularly the positional changes of large objects, significantly altering the scene in MOR scenarios. Consequently, deploying SLAM-based methods in our tasks proves challenging due to these dynamic and evolving environmental conditions.
>
>
> 2. “The proposed method heavily relies on the localization network (L-Net). However, training the L-Net necessitates extensive pre-training (e.g., 10,000 episodes in the proposed simplified environment), raising questions about its suitability for deployment in more complex environments like Habitat, AI2Thor, or the real world.”
>
> Regarding the applicability of L-Net in more photorealistic environments, such as Habitats or AI2Thor, we contend that its deployment in these simulators should not pose significant challenges. Notably, simulators like Habitat[1] are efficient, achieving over 1,000 steps per second (SPS), enabling the completion of an entire episode in approximately one second. Consequently, the collection of substantial datasets within the Habitat simulator can be accomplished in a matter of hours.
>
> Comparatively, for training L-Net, we rely on a real-world dataset comprising only 20,000 steps. Acquiring this amount of data would typically take less than a day. This efficiency underscores the suitability of L-Net for real-world setups.
>
> [1] Szot, Andrew, et al. "Habitat 2.0: Training home assistants to rearrange their habitat." NIPS 2021
>
> 3. “The evaluation of the proposed method takes place in toy-like environments, and real-world experiments assume ground-truth poses of agents and objects.”
>
> We agree that the evaluation of baselines and our method is in the simplified environment, but dynamic involved in this simulation environment such as wall texture, the types and colors of each object changed every episode, and the introduced motion uncertainty makes MOR still non-trivial for baselines and even challenging for proposed method according to the experiment results.
>
> Presently, our focus extends to real-world experiments, where the agent relies solely on RGB inputs for rearrangement tasks. We anticipate releasing these real-world experiment findings on our project webpage shortly. The current version serves as a demonstration, employing ground truth (GT) information to showcase the setup of our real-world platform. This choice is made to exemplify the potential for transferring the trained policy seamlessly from simulation to the real-world setting without the need for any fine-tuning.
>
> 4. “I recommend that the authors enhance the clarity of the writing in Sections 3 and 4. These sections currently contain an overwhelming amount of details and lack structured and logically coherent expressions.”
>
> We thank the reviewer's valuable suggestion. The fundamental logic underlying Section 3 is to start with the definition of the MOR, treating it as a Partially Observable Markov Decision Process (POMDP). Subsequently, we decompose MOR into a series of subtasks. The section concludes by introducing evaluation metrics utilized in this paper to assess performance.
>
> In Section 4, the logical progression commences with the introduction of L-Net, followed by a detailed exploration of the policy net. The inclusion of these specifics aims to provide a comprehensive definition of the MOR task and elucidate the proposed method thoroughly.
>
> 5. “How does the agent perceive the target poses of the objects? Does it estimate these poses from another unshuffled environment?”
>
> The target pose of each object is directly given to the agent (in the global coordinate, similar as the task definition in Habitat[1]). So there is No need to estimate the target poses.
>
> [1] Szot, Andrew, et al. "Habitat 2.0: Training home assistants to rearrange their habitat." NIPS 2021

---

### Author Response · Authors · 2023-11-22
**To all reviewers: thank you all for the constructive feedbacks and encouragements.**

We sincerely appreciate all reviewers’ constructive feedback and positive comments: (1) “this paper studies an important and under explored problem which is the lack of perfect localization in the visual object rearrangement task” (Reviewer ZaeQ and CMyF); (2) “baselines and ablations for the rearrangement policy are well thought-out” (Reviewer XWmC); (3) “separate training of the pick and drop policies makes sense” (Reviewer CAZo).

According to the reviewers’ comments, we further improve the quality of the paper and highlight all the changes of the manuscript with blue text. Then, we address each reviewer’s questions individually in the following sections.

---

### Meta-Review · Area_Chair_a3zo · 2023-12-03

**Metareview:**

The paper received reject ratings from all reviewers (5,5,3,3). The reviewers had several concerns such as using toy-like environments, missing baselines for localization, and L-Net being an overkill for the proposed task. While the rebuttal addressed some of the concerns, the main issue is that the environment is visually and structurally trivial. Given the state of the field and various rearrangement works in more complex environments such as Habitat and AI2-THOR, the AC believes that the paper does not meet the bar for publication at ICLR. Therefore, the AC follows the recommendation of the reviewers and recommends rejection.

**Justification For Why Not Higher Score:**

The main issue is that the environments used for the experiments are quite simplistic, the proposed approach will struggle for more complex environments.

**Justification For Why Not Lower Score:**

N/A

---

### Decision · Program_Chairs · 2024-01-16

Reject